# A Multi-Omics-Based Exploration of the Predictive Role of MSMB in Prostate Cancer Recurrence: A Study Using Bayesian Inverse Convolution and 10 Machine Learning Combinations

**DOI:** 10.3390/biomedicines13020487

**Published:** 2025-02-16

**Authors:** Shan Huang, Hang Yin

**Affiliations:** Department of Urology, Beijing Chao-Yang Hospital, Capital Medical University, Beijing 100020, China; 18661656191@163.com

**Keywords:** prostate cancer, biochemical recurrence, Bayesian deconvolution, machine learning

## Abstract

**Background:** Prostate cancer (PCa) is a prevalent malignancy among elderly men. Biochemical recurrence (BCR), which typically occurs after radical treatments such as radical prostatectomy or radiation therapy, serves as a critical indicator of potential disease progression. However, reliable and effective methods for predicting BCR in PCa patients remain limited. **Methods:** In this study, we used Bayesian deconvolution combined with 10 machine learning algorithms to build a five-gene model for predicting PCa progression. The model and the five selected genes were externally validated. Various analyses such as prognosis, clinical subgroups, tumor microenvironment, immunity, genetic variants, and drug sensitivity were performed on MSMB/Epithelial_cells subgroups. **Results**: Our model outperformed 102 previously published prognostic features. Notably, PCa patients with a high proportion of MSMB/epithelial cells were characterized by a greater progression-free Interval (PFI), a higher proportion of early-stage tumors, a lower stromal component, and a reduced presence of tumor-associated fibroblasts (CAF). The high proportion of MSMB/epithelial cells was also associated with higher frequencies of SPOP and TP53 mutations. Drug sensitivity analysis revealed that patients with a poorer prognosis and lower MSMB/epithelial cell ratio showed increased sensitivity to cyclophosphamide, cisplatin, and dasatinib. **Conclusions**: The model developed in this study provides a robust and accurate tool for predicting PCa progression. It offers significant potential for enhancing risk stratification and informing personalized treatment strategies for PCa patients.

## 1. Introduction

Prostate cancer (PCa), a malignant tumor that is androgen-dependent [1], is the second most common cancer and the fifth leading cause of cancer-related mortality in males [2,3]. The gold standard for diagnosis is a prostate biopsy, which can be complemented by the patient’s clinical manifestations, digital rectal examination, prostate-specific antigen (PSA) levels, and imaging to further define the patient’s condition. The treatment plan is determined based on the tumor stage, the patient’s overall health, and the specific characteristics of the tumor. For early-stage localized PCa, radical prostatectomy (RP) [4] and radiotherapy (RT) [5] are the primary treatment options, offering significant benefits. For advanced or metastatic prostate cancer, androgen deprivation therapy is commonly used, with chemotherapy, immunotherapy, and targeted therapies available in later stages. Approximately 20–40% of patients undergoing RP [6,7] and 30–50% of men treated with RT experience BCR within 10 years post-treatment [8,9]. The traditional definition of BCR is based on two criteria: (1) an increase in serum prostate-specific antigen (PSA) levels to ≥0.2 ng/mL on two consecutive measurements after RP, and (2) an increase in PSA levels to ≥2 ng/mL following RT [10,11,12,13]. However, predicting the risk of BCR after radical treatment for PCa patients remains challenging, often leading to either undertreatment or overtreatment.

PSA remains the standard method for monitoring BCR in PCa patients after treatment. However, it is not effective for the early prediction of BCR, and PSA testing may lead to overtreatment due to its high false-positive rate [14]. The advancement of sequencing technologies has provided significant technical support for the development of cancer prognosis models [15]. Moreover, artificial intelligence and machine learning, through training on vast amounts of data, have made considerable success in guiding decision-making in the medical field [16]. Therefore, developing a more accurate and reliable model for predicting BCR in PCa patients is crucial.

In this study, we employed Bayesian deconvolution to predict cell types in PCa from RNA sequencing data [17]. We compiled clinical data and RNA sequencing data from multiple PCa patients and constructed a prognostic model for post-treatment BCR (or DFI) using a combination of 10 machine learning approaches, which included 117 algorithms. This model aims to guide risk stratification and personalized treatment of PCa patients.

## 2. Methods

### 2.1. Study Design

The study collected and analyzed five clinical cohorts of PCa and, based on these cohorts, constructed and validated a predictive model containing five genes for predicting DFI (occurrence of BCR) in patients with PCa using a combination of 117 algorithms.

### 2.2. Data Collection and Preprocessing

We retrospectively collected data from multiple global centers on PCa cohorts and selected samples for this study based on the following criteria: (1) primary PCa, (2) availability of RNA sequencing data, (3) postoperative follow-up time of at least 30 days, and (4) patients who had undergone RP or RT. Ultimately, we identified five cohorts: The Cancer Genome Atlas (TCGA-PRAD) [18], GSE70769, GSE70768, GSE116918, and GSE54460. RNA sequencing and clinical data for TCGA-PRAD were downloaded and organized using the “TCGAbiolinks” package, and DFI data were obtained from the Xena website (https://xena.ucsc.edu/, accessed on 10 December 2024). The remaining four cohorts were downloaded from the GEO database (https://www.ncbi.nlm.nih.gov/geo/, accessed on 10 December 2024) [19]. Gene expression levels in all datasets were represented on a log2 scale. The TCGA-PRAD data includes complete clinical and RNA sequencing data for 500 PCa patients. It provides primary tumor and metastatic tissue samples from patients aged 40 to 80 years. Inclusion criteria: patients diagnosed with PCa between 2000 and 2015; complete RNA sequencing and clinical follow-up data available. Exclusion criteria: patients with incomplete RNA sequencing or clinical follow-up data. The GEO data were used as a validation dataset, and all patients with a diagnosis of PCa were included based on the inclusion criteria.

The single-cell dataset (GSE221603) was sourced from the GEO database. We used the Seurat package [20] (v4.3.3) to preprocess this data, initially excluding cells with fewer than 200 or more than 10,000 genes, as well as those with mitochondrial gene proportions greater than 20% and ribosomal gene proportions greater than 20%, as these cells are considered low quality. Mitochondrial gene proportions serve as an indicator of cell health; an increase in mitochondrial gene expression typically occurs when cells are under stress or undergoing cell death. Therefore, excluding cells with high mitochondrial gene proportions helps ensure the inclusion of high-quality cells. A threshold of 20% is commonly used in the literature [21,22]. Using a lower threshold might retain more cells but fail to exclude low-quality ones, while a higher threshold might exclude too many cells and lose important subpopulations. After this, the “NormalizeData” function was applied to normalize the processed data using “LogNormalize”. We then identified the top 2000 highly variable genes for downstream analysis using the “FindVariableFeatures” function. The data were subsequently scaled using the “ScaleData” function to ensure that gene expression values had a mean of zero and a variance of one [23]. Dimensionality reduction was performed using t-distributed Stochastic Neighbor Embedding (tSNE) and Uniform Manifold Approximation and Projection (UMAP), followed by cluster analysis using the “FindNeighbors” and “FindClusters” functions. The results were used for further single-cell and spatial transcriptomics analyses. The spatial transcriptomics data for sample VISDS000565 were downloaded from the CROST database [24] (https://ngdc.cncb.ac.cn/crost/, (accessed on 10 December 2024), including the expression matrix in H5 format and the corresponding Spatial folder.

Expression quantitative trait loci (eQTL) are genomic regions that specifically regulate mRNA and protein expression levels, with these levels being proportional to quantitative traits. Protein quantitative trait loci (pQTL) are genetic variants associated with protein expression levels, which can affect gene transcription, translation, and other processes. In this study, we conducted a summary-data-based Mendelian randomization (SMR) analysis for PCa using aggregated eQTL and pQTL data. The SMR analysis was performed using the SMR Portal [25] (westlake.edu.cn, accessed on 5 December 2024). The eQTL data were sourced from eQTLGen (blood tissue, *n* = 31,684) and GTEx v8 eQTL (prostate tissue, n = 245; blood tissue, n = 755). The pQTL data were derived from three distinct cohorts: pQTL_INTERVAL (n = 3301), pQTL_FENLAND (n = 10,708), and pQTL_SCALLOP (n = 30,931). Prostate cancer data were obtained from the public database IEU OPEN GWAS (https://gwas.mrcieu.ac.uk/datasets/, accessed on 5 December 2024). We selected the PCa data with GWAS ID ieu-b-85, downloaded the VCF file, and organized the data. This dataset, originating from the PRACTICAL website [26] (http://practical.icr.ac.uk/blog/, accessed on 5 December 2024), includes 140,254 individuals (79,148 cases and 61,106 controls), all of European descent. We used the organized data as the outcome factor for subsequent analysis.

### 2.3. SMR Analysis and Single-Cell Analysis

SMR analysis is capable of identifying genetic loci that have a causal association with outcome traits at the genetic level [23]. We performed SMR analysis on the curated PCa data using summary statistics of eQTL and pQTL to obtain pQTL data with a causal relationship to PCa. A *p*_SMR value < 0.05 and a *p*_HEIDI value > 0.01 were considered indicative of significant causal associations.

For the preprocessed single-cell data, we used the SingleR package [27] to classify cell clusters into different cell types. Subsequently, we applied the “Cellchat” package [28,29] and the “Vector” package [30] to perform cell communication analysis and pseudotime analysis, respectively, in order to explore the interactions between different cell types and the evolutionary process of cellular subtypes during development.

### 2.4. Spatial Transcriptome Analysis

The scRNA-seq data were obtained from sample GSM6890196 in GSE221306, and the spatial transcriptome data were sourced from the CROST database mentioned above. In this study, we focused on analyzing the spatial expression patterns of MSMB in epithelial cells. To achieve this, we categorized the epithelial cells based on MSMB expression levels into MSMB-positive and MSMB-negative groups. We then performed cell communication analysis and spatial transcriptomics for these subgroups, constructed homotypic and heterotypic cell networks, and described the co-localization of cell types using the “mistyR” package [31].

### 2.5. BayesPrism Inverse Convolution Analysis

The BayesPrism package can be downloaded and installed from the GitHub repository (https://github.com/Danko-Lab/BayesPrism, accessed on 15 December 2024). BayesPrism infers tumor microenvironment composition and gene expression from bulkRNA data based on scRNA data [17]. We further subdivided the obtained cell types again according to the results of the SMR analysis, cell cluster marker genes, and cell type marker genes to obtain more refined immune cell subpopulations. BayesPrism was then applied to analyze the correlation between these subdivided immune cell subpopulations and the clinical information of PCa, as well as their response to immunotherapy, drug prediction, and single nucleotide variant (SNV) analysis.

### 2.6. Weighted Gene Co-Expression Network Analysis (WGCNA) of Convoluted Cells

WGCNA, a multifactorial association analysis that integrates genes, samples, target modules, and traits, was used in this study to identify modular genes that are highly associated with convoluted cells [32]. First, excessive missing values and outlier samples were examined and removed; the remaining samples were then clustered to eliminate abnormal ones. Next, we tuned the soft threshold to select the optimal power based on the scale-free topology model fit (R^2^) and the mean connectivity. The optimal power was chosen when the scale-free topology model fit reached its highest value, and the mean connectivity leveled off, reaching its lowest value. The co-expression network was then constructed and divided into modules based on the selected power values. Gene significance (GS) indicates the correlation between genes and clinical phenotypes, while module significance (MS) reflects the relationship between modules and disease states [33].

### 2.7. Construction of PCa DFI Model Based on Machine Learning

First, we selected the marker genes of the chosen cell types from single-cell analysis (Markers), trait-related genes obtained from Pearson correlation analysis (PCC), differentially expressed genes from TCGA (Diff_gene), and module genes with the highest correlation to convoluted cells identified by WGCNA as intersections to obtain 44 significant genes. Using gene expression matrices from the five clinical cohorts mentioned above, we finally selected 28 core genes for constructing the PCa DFI prognostic model. The “Mime” package (https://github.com/l-magnificence/Mime, accessed on 23 December 2024) was used to screen five prognosis-related genes through univariate Cox regression. Then, we applied 10 machine learning algorithms (Randomized Survival Forest (RSF) [34], Survival-Support Vector Machine (Survival-SVM) [35], Least Absolute Shrinkage and Selection Operator (Lasso), Stepwise Cox, Ridge Regression, Generalized Boosted Regression Modeling (GBM), Elastic Networks (Enet), CoxBoost, Partial least squares regression for Cox (plsRcox) [36], and Supervised Principal Components (SuperPC)) in combination on the TCGA-PRAD cohort for training and validated the model in the GSE70769 cohort [37]. The best combination of algorithms was selected based on the median C-index value.

### 2.8. Acquisition of Published Signatures

The predictive performance of the PCa DFI model was further validated by comparing it with a total of 120 PCa prognostic signatures screened by Li, Yin et al. [38,39]. The cindex_comp and auc_comp functions in the “Mime” package were used to compare the C-index and 1-, 3- and 5-year AUC values of our model with those of the other 102 models.

### 2.9. Enrichment Analysis

We performed Gene Ontology (GO) [40], Kyoto Encyclopedia of Genes and Genomes (KEGG) [41], and Disease Ontology (DO) [42] analyses on 44 important genes to identify functions, pathways, and diseases that were significantly enriched in the gene sets. Additionally, Gene Set Enrichment Analysis (GSEA) was conducted on the five prognostic genes to identify significantly enriched functional pathways. A corrected *p*-value of less than 0.05 was considered statistically significant.

### 2.10. Prognostic Analysis of Prognostic Genes

Based on the gene expression matrix of the TCGA-PRAD cohort and clinical information on overall survival (OS), DFI, disease-specific survival (DSS), and progression-free survival (PFS), we analyzed the expression of five prognostic genes for survival analysis of OS, DFI, DSS, and PFS to further validate the predictive ability of these genes in PCa.

### 2.11. Correlation Analysis Between Prognosis-Related Genes and Convoluted Cells

To verify the association between the final screened prognosis-related genes and specific convoluted cells, we analyzed the correlation between these genes and convoluted cell subpopulations in the TCGA-PRAD cohort using the Spearman method.

### 2.12. Statistical Analysis

Statistical analysis and visualization were performed using R software (version 4.4.0). Spearman correlation analysis was used to assess the relationship between prognostic genes and convoluted cells. The Wilcoxon rank-sum test was applied to analyze the correlation between specific convoluted cells and the tumor microenvironment, as well as the set of immune-related genes. Ten types of machine learning models were performed using the Mime package, and the predictive signatures of multiple models were compared. *p*-values less than 0.05 were considered statistically significant.

## 3. Results

### 3.1. SMR Analysis

In this study, proteins related to PCa risk were first identified through SMR analysis. Appendix A show the results of the SMR analysis. The eQTL analysis indicated that increased mRNA levels of CASP8, CTBP2, NCOA4, ZBTB38, NUCKS1, and SLC25A37 were associated with an increased risk of PCa, while decreased mRNA levels of WTAP, SNPPC, and TCF7L2 were also linked to an increased risk of PCa. The pQTL analysis revealed that elevated protein abundance of EHBP1 and SPINT2 was significantly associated with a higher risk of PCa, while decreased protein levels of MSMB, CTSS, PRSS3, and ARL3 were significantly associated with an increased risk of PCa. This suggests that MSMB, CTSS, PRSS3, and ARL3 proteins may act as protective factors against PCa.

### 3.2. Verify the Expression Characteristics of Gene MSMB in PCa by Single-Cell and Spatial Transcriptome

We analyzed the expression specificity and differences of MSMB in normal tissue versus PCa, and normal tissue versus metastatic hormone-sensitive prostate cancer (mHSPC), respectively. In the single-cell analysis of normal tissue versus PCa, we classified 17 cell clusters into 7 cell types (Figure 1A); in the single-cell analysis of normal tissue versus mHSPC, we classified 14 cell clusters into 7 cell types (Figure 1B). Based on the marker genes of different cell types, we found that MSMB was predominantly expressed in epithelial cells (MSMB/Epithelial_cells) (Figure 1C). MSMB/Epithelial_cells were further categorized into high- and low-expression groups based on the median MSMB expression level in epithelial cells. The results showed that the proportion of high MSMB expression in MSMB/Epithelial_cells progressively decreased with cancer progression (Figure 1D). KEGG and GSEA analyses revealed that the differentially expressed genes in both high- and low-expression groups were enriched in the oxidative phosphorylation and reactive oxygen species pathways, which were upregulated in the high-expression group in both PCa and mHSPC analyses (Figure 1E). Compared to normal tissues, the number and strength of cellular communications were reduced in PCa, with further attenuation observed as the cancer progressed (Figure 1F). The communication activities between cellular subpopulations in PCa are shown in Figure 1G. Stronger cellular communication activities were observed between WTAP/Macrophage and iPS_cells versus MSMB/Epithelial_cells; the ligand-receptor interactions between the two are shown in Figure 1H. Pseudotime analysis revealed that MSMB/Epithelial_cells were positioned at the developmental endpoint (Figure 1I).

Spatial transcriptome analysis of the subdivided cell subpopulations was performed using RCTD reverse convolution. We observed that MSMB+Epithelial_cells exhibited a spatial distribution similar to that of iPS_cells (Figure 2A). To analyze how MSMB+Epithelial_cells are influenced by surrounding environmental cells in space, we constructed homotypic and heterotypic cell networks. Figure 2B shows concentrated areas of interactions between MSMB+Epithelial_cells and their own cell type, while Figure 2C illustrates interactions between MSMB+Epithelial_cells and other cell types, further confirmed by enrichment scores of these interactions. Co-localization analysis of various cell types was performed using the “mistyR” package. We defined three spatial scales: intra (0 spot radius), juxta_5 (5 spot radius), and para_15 (15 spot radius). The results revealed that the spatial dependence between MSMB+Epithelial_cells and iPS_cells was strongest at the point scale and gradually weakened as the distance increased (Figure 2D).

### 3.3. scPagwas Screening for Trait-Associated Genes and Cell Populations

The scPagwas computational method developed by Ma et al. integrates pathway activation transformations of scRNA-seq data and GWAS summary statistics to reveal the cellular context associated with traits, effectively reducing the technical noise and high sparsity inherent in scRNA-seq data [43]. scPagwas was used to analyze the trait-relevant score (TRS) of different cell types, and it was found that the epithelial cell population, T cell population, and monocyte population had higher TRS scores. The TRS of MSMB/Epithelial_cells differed from that of the remaining cells, except for MSMB/Macrophage (Figure 3A). Figure 3B shows the top 10 genes significantly associated with PCa. Subsequently, significant cell types associated with the traits were inferred using the self-help method included in the scPagwas package. As shown in Figure 3C and Figure 3D, ZBTB3/Tissue_stem_cells and MSMB/Macrophage were significantly associated with the traits.

### 3.4. Bayesian Inverse Convolution Analysis of the Prognostic Properties of Convoluted Cells

We plotted a heatmap to explore the correlation between cell types and determine if there is a high correlation between different cell subpopulations. The results showed a strong correlation between MSMB/Epithelial_cells and iPS_cells (Figure 4A). Subsequently, we examined the expression consistency of different types of genes. Figure 4B visualizes abnormal genes (e.g., ribosomal protein genes, mitochondrial genes, etc.) in the scRNA data. It can be seen that these genes exhibit higher average expression and lower cell-type specificity scores and were, therefore, excluded because they could interfere with clustering and bias the results. Figure 4C shows the highest correlation for protein-coding gene groups. To reduce the batch effect and speed up the computation, we primarily deconvolved the protein-coding genes [23,44]. In this study, we correlated the convoluted cells with the prognostic clinical information of PCa patients in the TCGA-PRAD cohort. Four samples with follow-up times ≤30 days were excluded, and survival curves for OS, DFI, DSS, and PFS were obtained for each convoluted cell type in PCa patients. Figure 4D shows the survival analysis of cell types significantly correlated with prognosis (*p* < 0.05). The analysis reveals that PCa patients with different cell subtypes have distinct survival outcomes. iPS cells are significantly correlated with DFI: the higher the proportion of iPS cells, the lower the DFI, indicating a significant negative correlation between the proportion of iPS cells and DFI. MSMB/Epithelial_cells are significantly correlated with PFI: the higher the proportion of these cells, the greater the PFI, indicating a significant positive correlation between the proportion of MSMB/Epithelial_cells and PFI. Thus, MSMB/Epithelial_cells may serve as a potential protective factor for the prognosis of PCa patients, further confirming the protective effect of MSMB proteins against PCa observed in the SMR analysis.

### 3.5. Clinical Subgroup Analysis of Convolutional Cells

For different Bayesian inverse convolutional cells, we analyzed the survival prognosis differences across various clinical parameter subtypes. The results suggested that different convolutional cells showed distinct survival prognoses in clinical subgroups. MSMB/Epithelial_cells differed significantly between T1 and T2, N0 and N1, and M0 and M1 stages; MSMB/Macrophage showed differences between patients aged ≤65 years and >65 years (Figure 5A). These findings indicate that the proportion of cell types characterized by MSMB expression is higher at earlier stages or in younger ages, which may suggest that a high proportion of MSMB/Epithelial_cells and MSMB/Macrophage is associated with a better prognosis.

### 3.6. Correlation Analysis of MSMB/Macrophage with the Remaining Convoluted Cells and Tumor Microenvironment (TME)

Based on the significant cell types associated with traits obtained by the self-help method of scPagwas analysis, we further selected MSMB/Macrophage for subsequent analysis. Tumor stromal cells and immune cells also play an important role in tumor prognosis. In order to comprehensively assess the tumor microenvironment, we applied the “tidyestimate” package to calculate the stromal score, immune score, ESTIMATE score and inferred tumor purity between the MSMB/Macrophage high- and low-ratio groups in prostate cancer. At the same time, we also analyzed the differential analysis between the MSMB/Macrophage high-low-ratio group and the rest of the convoluted cell types. The results are shown in Figure 5B,C, the proportion of MSMB/Macrophage with iPS_cells, WTAP/Macrophage, Endothelial_cells, NCOA4/Monocyte, TCF7L2/Epithelial_cells, ZBTB38/Tissue_stem_cells, SNRPC/Monocyte, and SLC25A37/Monocytede were significantly negatively correlated; and significantly positively correlated with the proportions of MSMB/Epithelial_cells and CMP. Tumor microenvironment analysis showed that a high proportion of MSMB/Macrophage was significantly correlated with a lower stromal component, which was closely associated with tumor progression, which may imply that a high proportion of MSMB/Macrophage is associated with a better prognosis. The volcano plot shows the more significant differential genes between the high- and low-ratio groups. Similarly, the results for MSMB_Epithelial_cells showed that a high proportion of this cell type was also significantly associated with a lower stromal component. Differential analysis of the remaining convoluted cells can be seen in Appendix A.

### 3.7. Immunocorrelation Analysis of Convoluted Cells

The set of immune-related genes was downloaded from the immport database (https://www.immport.org/shared/genelists, accessed on 20 December 2024). Figure 6A shows that SYK genes in the BCR signaling pathway were significantly different between the high and low percentage subgroups of MSMB/Epithelial_cells, with lower expression observed in the high percentage group [45]. Immunoanalysis of chemokines and chemokine receptors revealed that chemokines and their receptor genes were predominantly lowly expressed in the high percentage MSMB/Epithelial_cells subgroup (Figure 6B,C), which may be associated with a better prognosis [46,47,48,49,50]. Analysis of cytokines and their receptors indicated that the high-proportion group had lower levels of IL-6, IL-10, IL-17, and others (Appendix A), suggesting that PCa progression may be attenuated in this group compared to the low-proportion group [51]. The expression levels of TGFb family members were mostly lower in the high-proportion group than in the low-proportion group of MSMB/Epithelial_cells (Figure 6D,E), which may help inhibit PCa progression [52].

Immunotherapy analysis showed that the levels of myeloid-derived suppressor cells (MDSC), cancer-associated fibroblasts (CAF), CD274 (PD-L1), and immune exclusion were lower in the high-ratio group compared to the low-ratio MSMB/Epithelial_cells group (Figure 7A). As an important component of TME [53], MDSC are a heterogeneous class of cells that inhibit T cell function through multiple pathways [54,55]. CAF are crucial in promoting tumor growth and metastasis, making them a potential therapeutic target for PCa [56]. CD247, also known as PD-L1, is widely expressed in various cancer cells and can impair T cell-mediated immune responses [57]. The immune exclusion score assesses the degree of TME rejection of immune cells. All four metrics were lower in the high-proportion MSMB/Epithelial_cells group, suggesting that this group is associated with a favorable prognosis (MSMB/Macrophage).

In this study, differences in drug IC50 values between the high- and low-ratio groups were analyzed using the oncoPredict package and the Genome of Drug Sensitivity in Cancer (GDSC) databases (Figure 7B). The results indicate that the low percentage MSMB/Epithelial_cells group has higher sensitivity to drugs such as cyclophosphamide, cisplatin, and dasatinib (Figure 7C), which may aid in personalizing treatment for specific PCa types.

### 3.8. Unique Genetic Variants Associated with MSMB/Epithelial_Cells Subgroups

Genetic characterization of the TCGA-PRAD cohort, including somatic mutation and copy number variation (CNV) data, is available from the Genome Data Commons (GDC, https://portal.gdc.cancer.gov/, accessed on 20 December 2024). In this study, the downloaded SNV data were analyzed using the R package “maftools” [58]. We visualized the data in MAF format with the “maftools” package. The summary information of the MAF file was plotted using the function “plotmafSummary” (Figure 8A). The most common variants were missense mutations, and the most frequent variant type was single nucleotide polymorphisms (SNPs). The data also indicated that cytosine (C) to thymine (T) mutations were the most common. The median number of variants per sample was 21. Additionally, we listed the top 10 mutated genes, with mutations in TTN, TP53, and SPOP being the most frequent. Using the *oncoplots* package, we created a waterfall plot for the top 30 mutated genes (Figure 8B), where SPOP and TP53 showed the highest mutation rates (11%), followed by TTN mutations (9%). The predominant mutation type was missense mutations. Figure 8C,D shows the difference in mutation frequencies between the high and low percentage groups of MSMB/Epithelial_cells, with the high percentage group exhibiting a higher frequency of SPOP mutations and a lower frequency of TP53 mutations. SPOP mutations in the high-proportion group were primarily missense mutations, while TP53 mutations were mainly missense, frameshift, and splice-site mutations.

### 3.9. WGCNA Identification of Key Genes in Convoluted Cells

We performed WGCNA on convolutional cells to identify key genes for different convolutional cells. The optimal soft threshold (power) of 13 was selected based on R^2^ and mean connectivity, and a scale-independent topological network was constructed (Figure 9A). Module-convolutional cell correlation analysis showed that MSMB/Epithelial_cells and MSMB/Macrophage had the highest positive correlation with gray modules (Figure 9B). Scatter plots of correlation coefficients between MM and GS indicated that MSMB/Epithelial_cells and MSMB/Macrophage had significant positive MM-GS correlation with the gray module (Figure 9C). The genes in these two modules were selected for further analysis.

### 3.10. Construction and Evaluation of PCa DFI Prognostic Models

We identified 44 intersecting genes by taking the intersections of Markers, PCC, Diff_gene, and WGCNA (Figure 9D) and ultimately screened 28 genes across five clinical cohorts for subsequent analysis. First, we performed univariate Cox regression on the 28 core genes and selected five prognosis-related genes: “GGCT”, “C2”, “NDUFA11”, “MRPL41” and “SERHL2”. We verified the staining of “GGCT”, “C2” and “MRPL41” in PCa tissues using the Human Protein Atlas (HPA) database. Figure 9E shows that GGCT and MRPL41 were highly stained in PCa tissues, while C2 was not detected. Next, based on these genes, we constructed 117 prognostic models using TCGA-PRAD as a training set and validated them in the GSE70769 dataset. The C-index was calculated in both cohorts. We found that the StepCox[forward] + plsRcox model achieved the best results based on the ranking of the average C-index across all cohorts and the validation cohort (Figure 9F).

In model selection, we need to consider the balance between computational cost and model performance. For example, complex machine learning may require more computational time, while traditional models outperform complex algorithms in terms of computational cost. Therefore, we need to balance the issues of prediction performance and computational efficiency when selecting models in order to make a compromise between computation time, memory consumption, and prediction accuracy. To assess the prognostic value of the PCa DFI prognostic model, we divided the samples into high- and low-risk groups based on the median value. We then performed survival analyses to observe the differences between the two groups. The Kaplan–Meier survival analysis of StepCox[forward] + plsRcox in the TCGA-PRAD training set (N = 338) and the GSE70769 validation set (N = 92) showed statistically significant results (*p* < 0.05). Patients in the high-risk group had lower DFIs compared to those in the low-risk group, indicating that the high-risk group was more likely to experience earlier BCR (Figure 9G). The AUC values for the 117 machine learning models’ 1-, 3-, and 5-year survival probabilities are shown in Appendix A. It can be seen that StepCox[forward] + plsRcox is still able to achieve relatively superior performance in both cohorts. The average of the 2 cohort models’ 1-, 3-, and 5-year AUC values were 0.705, 0.765, and 0.765. Further validation of the model’s predictive power using time-dependent receiver operating characteristic (timeROC) curves. As shown in Figure 9H, the predicted AUC values for 1, 3, and 5 years in the TCGA-PRAD cohort were 0.694, 0.784, and 0.795, respectively; and the predicted AUC values for 1, 3, and 5 years in the GEO cohort (GSE70769) were 0.718, 0.745, and 0.731, respectively.

### 3.11. Comparison of PCa DFI Prognostic Models with 102 Published Models

The model has previously demonstrated strong predictive power and external generalization capabilities in predicting the presence or absence of progression in PCa. To further emphasize its prognostic value, we compared our model with 102 other models in terms of C-index and AUC values based on previous prognostic studies for multiple aspects of PCa (Appendix A). Figure 10A shows the different features ranked by C-index in the TCGA-PRAD and GSE70769 cohorts, with our model having a higher C-index than the other features. We also compare the predicted AUC values of these models for 1, 3, and 5 years and see that our model is slightly lower than Zhang.6.JO.36106334 for the first year AUC value in the TCGA-PRAD cohort (Figure 10B–D). Other than that, our model outperforms the rest of the signatures in all comparisons of predicted AUC values. All in all, our model maintains strong predictive power and external generalization ability compared to other signatures.

### 3.12. Enrichment Analysis

The results of GO enrichment analysis are shown in Figure 11A, where 44 significant genes were significantly associated with biological processes involving proteins and lipids. KEGG enrichment analysis indicated that these 44 significant genes were notably linked to pathways related to amino acid metabolism, lipid metabolism, and anti-tumor processes (Figure 11B). DO analysis revealed that these genes were significantly associated with Niemann-Pick Disease, a genetic disorder characterized by abnormal lipid metabolism (Figure 11C). All three analyses highlighted the enrichment of significant genes in metabolic abnormalities. We then performed single-gene GSEA on the five core prognostic genes. The enrichment results for the remaining four genes are shown in Figure 11D–G and Appendix A, except for MRPL41, which was not enriched in any pathway. The analysis revealed that all four genes were enriched in olfactory transduction. Additionally, the pathways that showed relative enrichment included porphyrin metabolism, ascorbate and aldarate metabolism, the proteasome, and cardiac muscle contraction. NDUFA11 was also enriched in the oxidative phosphorylation pathway.

### 3.13. Prognostic Analysis of Prognosis-Related Genes

In order to further verify the prognostic role of prognostic genes in PCa, we analyzed OS, DFI, PFI, and DSS for five genes, respectively, and the results are shown in Figure 12A–D. It can be seen that there were significant differences in DFI and PFI survival analysis in both high- and low-expression groups of C2 and NDUFA11, as well as in PFI in high- and low-expression groups of MRPL41. Among them, high expression of C2 was associated with a better prognosis, while high expression of NDUFA11 and MRPL41 was associated with a poor prognosis.

### 3.14. Relationship Between Prognosis-Related Genes and Convoluted Cells

The screening of prognostic genes is closely related to convoluted cells. In order to verify the relationship between prognostic-related genes and convoluted cells in the model, we used the Spearman method to correlate 5 core genes with 17 convoluted cells. The results suggested that a total of 57 results suggested a significant correlation between prognosis-related genes and convoluted cells (*p* < 0.05) (Appendix A); Figure 12E–H shows the images with significant correlation with MSMB/Epithelial_cells, and it can be found that except for NDUFA11, the other four core genes were significantly correlated with this convoluted cells

## 4. Discussion

PCa, being an indolent cancer, can often be cured early through surgery or radical radiotherapy. However, biochemical recurrence (BCR) serves as a critical turning point for patients with primary PCa after radical prostatectomy (RP), which can influence subsequent treatment strategies [59]. If not adequately managed, patients with BCR may progress to clinical recurrence [60]. Literature suggests that BCR often precedes clinical recurrence and serves as a predictor for an increased risk of distant metastasis, prostate cancer-specific mortality, and overall mortality [61]. Currently, the risk of BCR in patients with primary PCa after RP is mainly predicted based on clinical factors such as PSA levels, Gleason score, surgical margins, and lymph node metastasis [62]. However, there is still no better indicator to risk stratify PCa patients and predict BCR probability in advance.

This study focuses on the analysis of MSMB, as it is the strongest genome-wide predictor gene for PCa. Its protein levels are negatively correlated with overall, invasive, and early-onset PCa, and its protective role is particularly important for the disease, potentially providing a basis for PCa prevention strategies [63,64].

Currently, models for predicting tumors or non-tumors based on gene signatures are emerging rapidly, thanks to the rapid development of second-generation sequencing technology [65,66]. In this study, based on previous work and the characterized genes of MSMB/Epithelial_cells, we used 117 algorithm combinations consisting of 10 machine learning models to construct an optimal model for PCa DFI. The final model selected was a 5-gene prognostic signature constructed using the StepCox[forward] + plsRcox algorithm. Compared with 102 published PCa prognostic features, our model demonstrates clear advantages in both internal and external validation sets. Furthermore, the 5-gene model can help reduce the waste of medical resources to some extent.

Of the five prognostic genes screened, only GGCT was reported in PCa. Susumu Kageyama et al. compiled the expression characteristics and mechanisms of GGCT in cancer and found that GGCT is upregulated in various cancers, potentially playing a role in events leading to malignant phenotypes [67]. Yumiko Saito et al. found that knockdown of GGCT inhibited the growth of cancer cells in vitro and in vivo [68,69,70]. Bert Gold et al. found that C2 single nucleotide polymorphisms were associated with age-related macular degeneration (AMD) [71]. Suhg Namgoong et al. also found that the C2 gene contributes to the etiologic explanation of chronic hepatitis B in Koreans [72]. In terms of tumors, the C2 gene has been associated with non-Hodgkin’s lymphoma (NHL) [73] and colorectal cancer liver metastasis [74]. NDUFA11, as a Disulfidptosis gene, plays a role in the prognosis of diffuse large B-cell lymphoma (DLBCL) [75]. Wei Mao et al. found that NDUFA11 acts as a retinoid orphan nuclear receptor alpha (RORα) target, mediating its role in inhibiting superoxide anion generation in mitochondria, exerting inhibition of cancer metastasis and reducing macrophage accumulation [76]. It has also been suggested in the literature that NDUFA11 may be a potential therapeutic target in bladder cancer [77]. MRPL41 is a nuclear-encoded mitochondrial gene whose epigenetic regulation in breast cancer is influenced by estrogen receptor status [78]. Cheng Xin et al. experimentally verified that MRPL41 may play an inhibitory role in colorectal cancer progression [79]. Detailed information on prognosis-related genes can be found in Appendix A.

Our model was able to quantify the risk for PCa patients, categorizing them into high-risk and low-risk groups. We also performed a prognostic analysis of the core prognostic genes, finding that the C2 gene was positively correlated with DFI and PFI, the NDUFA11 gene was negatively correlated with DFI and PFI, and the MRPL41 gene was negatively correlated with DFI. Our results indicated that PCa patients with MSMB/Epithelial_cells in the low percentage group had a poorer prognosis. Therefore, we searched for highly sensitive drugs for the low percentage group to personalize treatment for high-risk patients. Ultimately, we found that the low percentage group, with its poorer prognosis, responded better to treatment with cyclophosphamide, cisplatin, and dasatinib compared to the high percentage group.

Our model has several advantages. First, we used multiple datasets with larger sample sizes than multiple other models, resulting in better generalization of the model. Second, many models use traditional models (e.g., Cox regression, etc.), which may not adequately capture nonlinear features in high-dimensional data, thus limiting predictive power. This study uses a larger sample size, including multiple datasets, which improves the generalizability and stability of the model. At the same time, 10 machine learning methods are used, which are able to capture the complex nonlinear relationships in the data and thus are more advantageous. In the future, we may further validate the model by employing a larger clinical cohort and may combine joint analysis with other fields (e.g., imaging histology, etc.) to improve model accuracy.

At the same time, our study has some limitations. First, the different sample sizes of the QTL datasets may lead to different statistical power in each study, which subsequently leads to errors in the results [80]. Second, the sample data collected were retrospective studies, and multicenter prospective studies are still needed to further validate the predictive value of our model. Third, due to incomplete clinical information in public databases, the potential relationships between variables could not be verified, and the conclusions might not be precise enough. Fourth, the molecular mechanisms of core prognostic genes in PCa progression remain unelucidated, and further ex vivo experiments are needed for validation.

## 5. Conclusions

In this study, we validated the prognostic role of MSMB/Epithelial_cells and MSMB/Macrophage in PCa patients for the first time based on Bayesian inverse convolution, single-cell and spatial transcriptomics, proteomics, and 10 types of machine learning and developed a stable and accurate signature based on specific genes of these two cells. This signature can help guide personalized treatment strategies, leading to improved prognosis for patients.

## Figures and Tables

**Figure 1 biomedicines-13-00487-f001:**
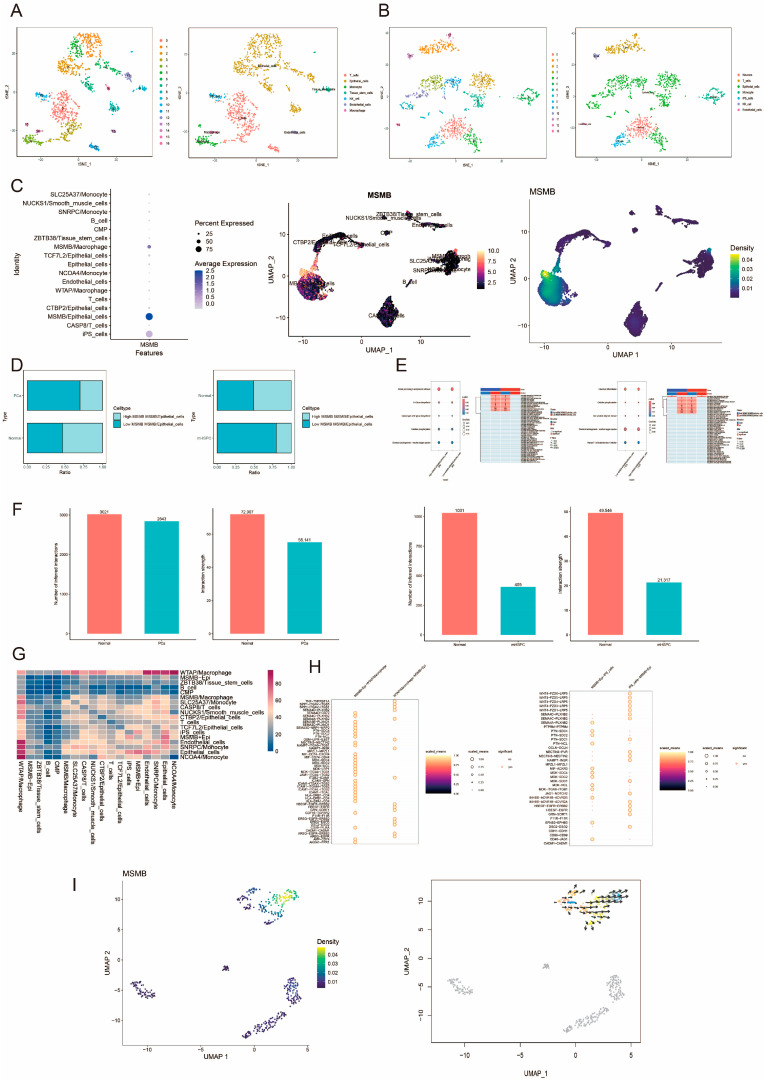
Single-cell analysis of MSMB in prostate cancer. (**A**) The tSNE plot of cell clusters and cell types in normal tissue versus PCa. (**B**) The tSNE plot of cell clusters and cell types of normal tissue with mHSPC. The different colored dots in figure (**A**,**B**) represent different cell clusters or cell types. (**C**) Expression levels of MSMB in different cells. The size of the circles in the diagram on the left indicates the proportion of gene expression. The larger the circle, the greater the proportion of gene expression. The color gradient indicates the strength of the gene expression level. Darker colors in the three graphs indicate higher MSMB expression. (**D**) Comparison of the differences between the high- and low-ratio groups of MSMB/Epithelial_cells in normal tissue versus PCa and normal tissue versus mHSPC. Light blue indicates MSMB/epithelial_cells high percentage group, and dark blue indicates MSMB/epithelial_cells low percentage group. (**E**) KEGG and GSEA analyses of normal tissues versus PCa and normal tissues versus mHSPC. The size of the circle in KEGG analysis represents the gene ratio. The color gradient indicates the size of the adjusted *p*-value, with darker colors representing smaller *p*-values. Blue in Cluster in GSEA analysis indicates a high-proportion group of MSMB/Epithelial_cells, and red indicates a low-proportion group of MSMB/Epithelial_cells. Blue in Direction indicates a downregulation, and red indicates an upregulation. In RRA, red color indicates statistical significance and vice versa. *: *p* < 0.05, **: *p* < 0.01, ***: *p* < 0.001, ****: *p* < 0.0001. (**F**) Comparison of the number and intensity of cellular communication between normal tissue and PCa, and normal tissue and mHSPC. (**G**) Comparison of the intensity of cellular communication activities between different cell subpopulations. The color gradient from blue to red indicates a gradual increase in the strength of the intercellular interaction. (**H**) Analysis of ligand-receptor interactions of MSMB_Epithelial_cells with WTAP/Macrophage and iPS_cells, respectively. Circle size and color gradient indicate the magnitude of ligand-receptor interactions. Larger circles and lighter colors indicate larger ligand-receptor interactions. A red line outside the circle indicates statistical significance and vice versa. (**I**) The pseudotime analysis of MSMB/Epithelial_cells. The color gradient indicates the level of gene expression density. Lighter colors indicate higher expression density.

**Figure 2 biomedicines-13-00487-f002:**
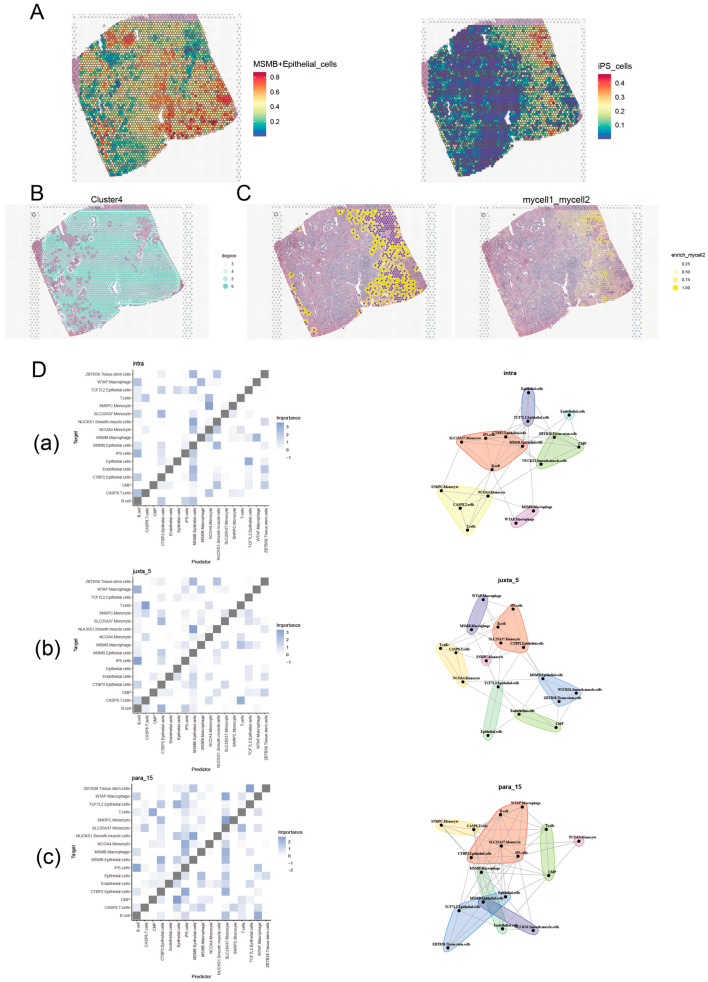
Spatial transcriptome analysis of MSMB/Epithelial_cells and iPS_cells. (**A**) Spatial expression characteristics of MSMB/Epithelial_cells and iPS_cells in prostate cancer. The color gradient represents the relative expression levels of MSMB. Blue to red color indicates low to high MSMB expression levels. (**B**) Homotypic cell network of MSMB/Epithelial_cells. The colors in the plot represent the degree of each spot, which refers to the number of neighboring spots that are spatially connected. Darker colors indicate spots with a higher degree of interaction, which can be indicative of more complex cellular interactions in that region. The white lines between spots represent spatial connections based on their proximity and network distance. (**C**) The spatial visualization of the interaction between MSMB/Epithelial_cells and iPS_cells in tissue samples. In the left image, the yellow points represent MSMB/epithelial cells, and the purple points represent iPS cells. The right panel shows the neighborhood enrichment score, quantifying the interaction strength between MSMB/epithelial_cells and iPS_cells. The color gradient reflects the enrichment level, with darker colors indicating higher enrichment and lighter colors indicating lower enrichment. (**D**) Co-expression analysis between cell subpopulations at different spatial scales. Figure (**a**–**c**) shows the analysis of different cell subpopulations at spatial scales of 0, 5, and 15, respectively. The figures shown in the left column are interaction heatmaps of cells at different spatial scales. The color gradient represents the strength of these interactions, with darker colors indicating stronger interactions. The images in the right column are community plots at different spatial scales. Each community represents a group of cells that are tightly distributed in space, and these cells may share functional similarities or collaborate in biological processes.

**Figure 3 biomedicines-13-00487-f003:**
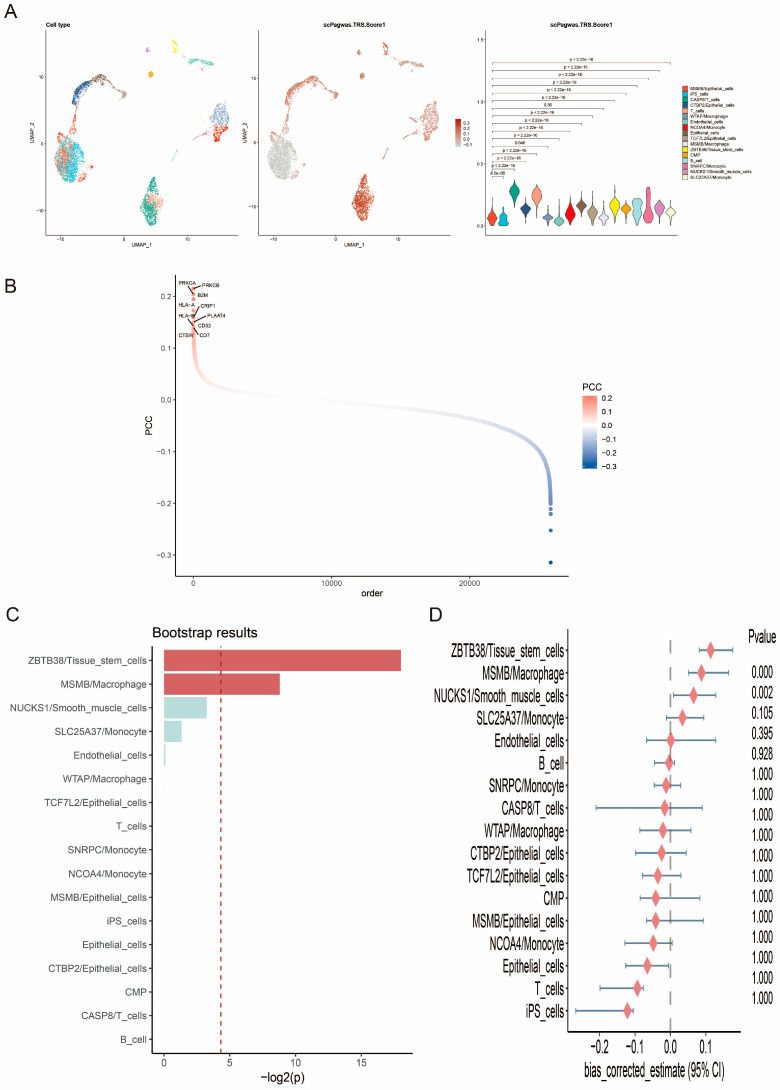
scPagwas screening for genes and cell populations associated with PCa. (**A**) TRS and differential analysis of different cell types. The color gradient in the second graph indicates the level of TRS scores; the darker the color, the higher the TRS score. The color gradient in the third graph indicates different cell types. (**B**) scPagwas identifies the top 10 genes significantly associated with PCa. Red color indicates positive correlation and blue color indicates negative correlation. (**C**) Cell types significantly associated with PCa inferred by self-help method. (**D**) Bias corrected estimate for each cell type inferred by self-help method.

**Figure 4 biomedicines-13-00487-f004:**
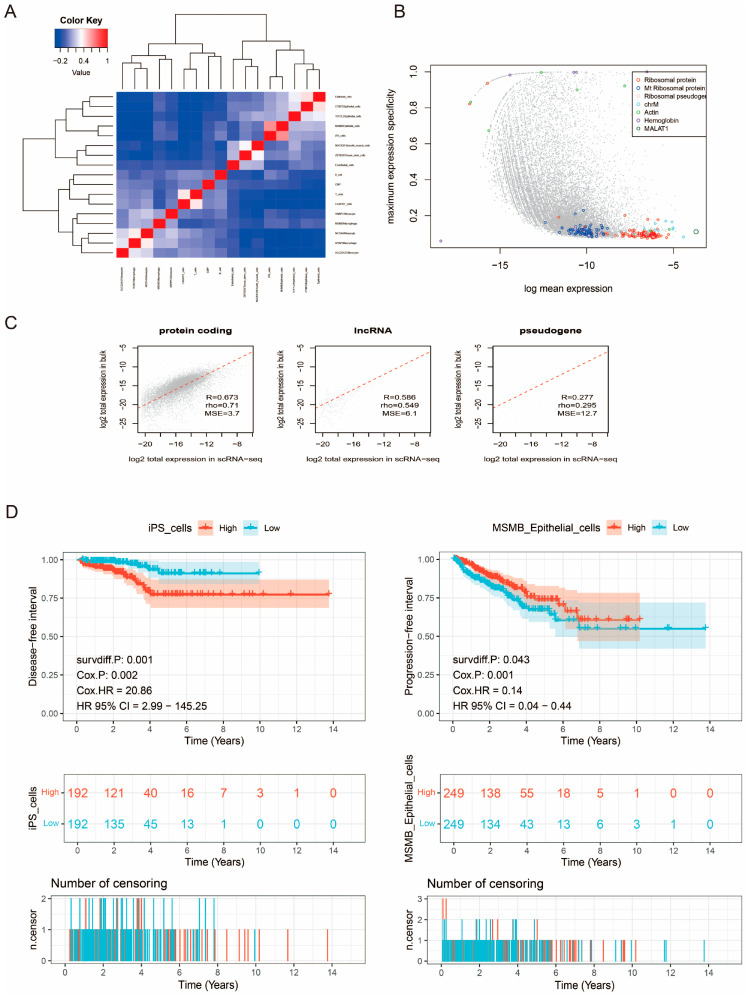
Bayesian inverse convolution analysis for different cell types. (**A**) Correlation analysis of different cell types. Red represents a positive correlation, and blue represents a negative correlation. The darker the color, the stronger the correlation. (**B**) Visualization of abnormal genes in scRNA data. Different colored circles represent different gene types. (**C**) Consistency test of different types of genes expressed in single-cell data and bulk data. A larger R value indicates higher consistency between single-cell data and bulk data. (**D**) DFI survival analysis of iPS cells and PFI survival analysis of MSMB_Epithelial_cells. Red curves indicate high-risk groups, and blue curves indicate low-risk groups. A *p*-value of less than 0.05 indicates statistical significance.

**Figure 5 biomedicines-13-00487-f005:**
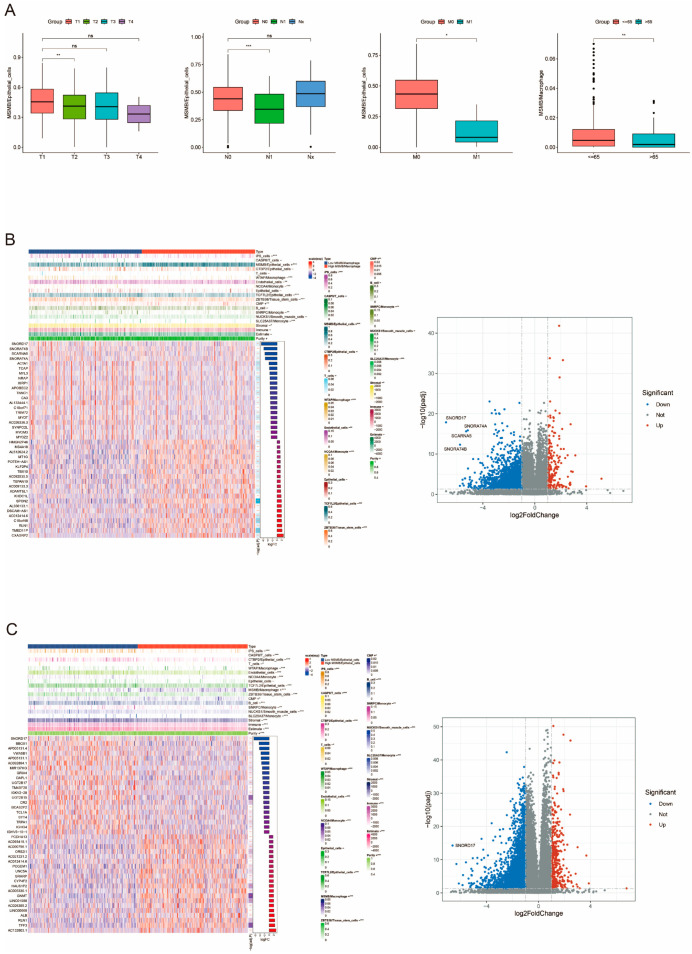
Multiple grouping analysis of convolutional cells. (**A**) Clinical subgroup analysis of MSMB/Epithelial_cells and MSMB/Macrophage. (**B**) Correlation analysis of MSMB/Macrophage with the rest of convoluted cells and tumor microenvironment analysis. The right panel shows the volcano plot of difference analysis between MSMB/Macrophage high and low percentage groups. (**C**) Correlation analysis of MSMB/Epithelial_cells with the rest of convoluted cells and tumor microenvironment analysis. The right panel demonstrates the volcano plot of difference analysis between MSMB/Epithelial_cells high and low percentage groups. The red color in Type in the heatmap indicates the high-expression group of cell types, and the blue color indicates the low-expression group. Different cell types and tumor microenvironment types are included in the notes on the right. The color gradient indicates the degree of cell type or tumor microenvironment type predominance. The darker the color, the higher its predominance. The blue color in the volcano plot indicates downregulation, and the red color indicates upregulation. * *p* < 0.05; ** *p* < 0.01; *** *p* < 0.001.

**Figure 6 biomedicines-13-00487-f006:**
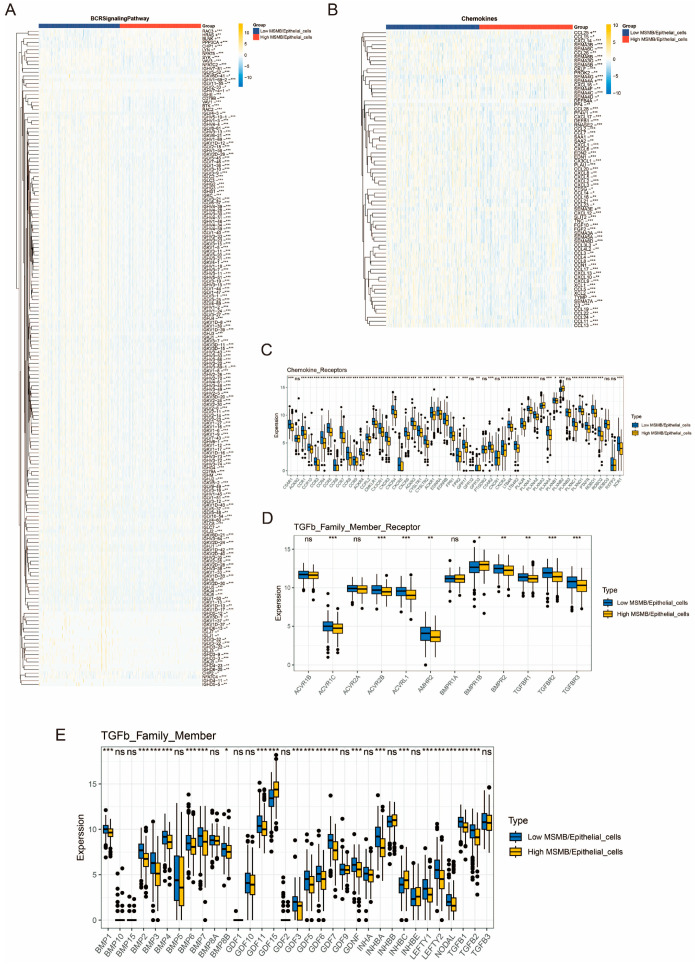
Analysis of immune-related gene sets in convoluted cells. (**A**) Differential analysis of BCR signal-ing pathway in MSMB/Epithelial_cells high- and low-ratio groups. Red indicates the high percentage group of MSMB/Epithelial_cells, while blue indicates the low percentage group of MSMB/Epithelial_cells. (**B**) Chemokine difference analysis of MSMB/Epithelial_cells high- and low-ratio groups. Red indicates the high percentage group of MSMB/Epithelial_cells, while blue indicates the low percentage group of MSMB/Epithelial_cells. The color gradient in Figure (**A**,**B**) indicates the expression levels of different genes, with yellow indicating high expression and blue indicating low expression. (**C**) Differential analysis of chemokine receptors in MSMB/Epithelial_cells high- and low-ratio groups. (**D**) Differential analysis of TGFb family member receptors in MSMB/Epithelial_cells high- and low-ratio groups. (**E**) Differential analysis of TGFb family members in MSMB/Epithelial_cells high- and low-ratio groups. Yellow color in Figure (**C**–**E**) indicates the high-expression group of MSMB/Epithelial_cells, while blue color indicates the low-expression group. ns: no significant difference; *: *p* < 0.05; **: *p* < 0.01; ***: *p* < 0.001.

**Figure 7 biomedicines-13-00487-f007:**
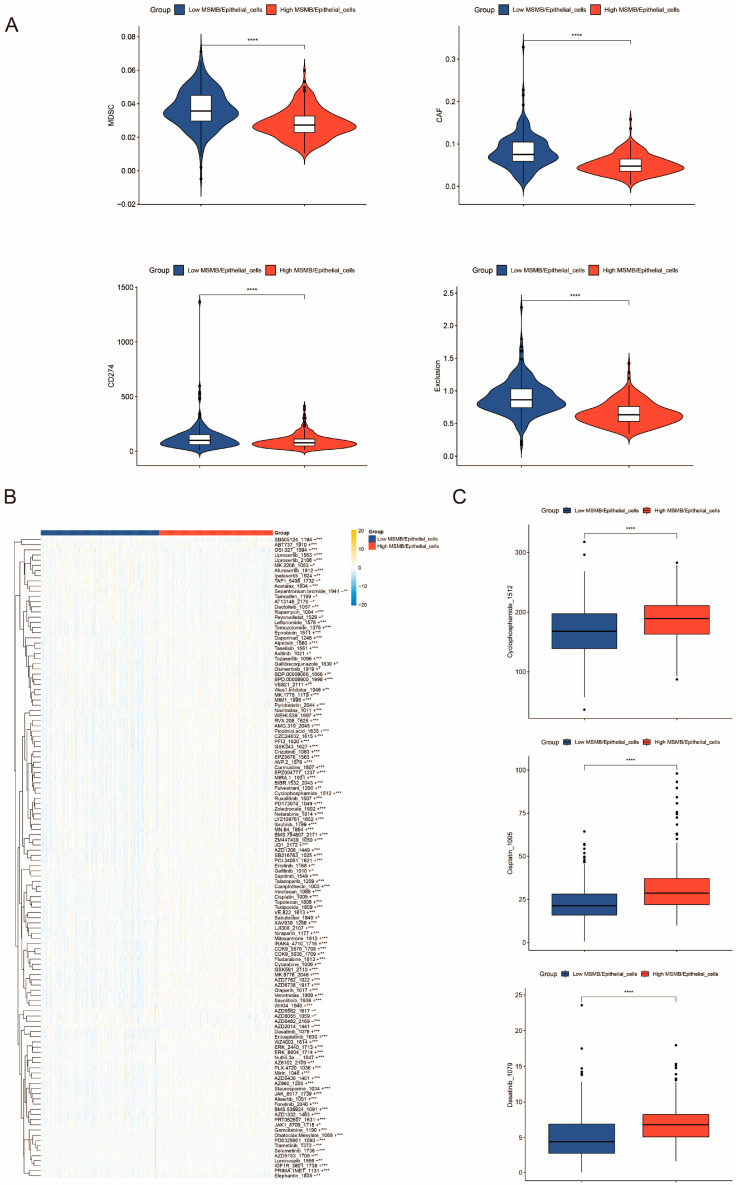
Immunotherapy and drug sensitivity analysis of convoluted cells. (**A**) Difference analysis between MDSC, CAF, CD274, and exclusion between MSMB/Epithelial_cells high- and low-ratio groups. (**B**) Drug sensitivity analysis between MSMB/Epithelial_cells high- and low-ratio groups. The color gradient represents the expression levels of different drugs. Yellow indicates high expression, while blue indicates low expression. “+” indicates that there is a positive correlation between the drug and the high percentage group and vice versa there is a negative correlation. (**C**) Box plots of the difference analysis of drug sensitivity of cyclophosphamide, cisplatin, and dasatinib between MSMB/Epithelial_cells high- and low-ratio groups. The red color in Figure (**A**–**C**) indicates the high percentage group of MSMB/Epithelial_cells, while blue indicates the low percentage group of MSMB/Epithelial_cells. * *p* < 0.05; ** *p* < 0.01; *** *p* < 0.001; **** *p* < 0.0001.

**Figure 8 biomedicines-13-00487-f008:**
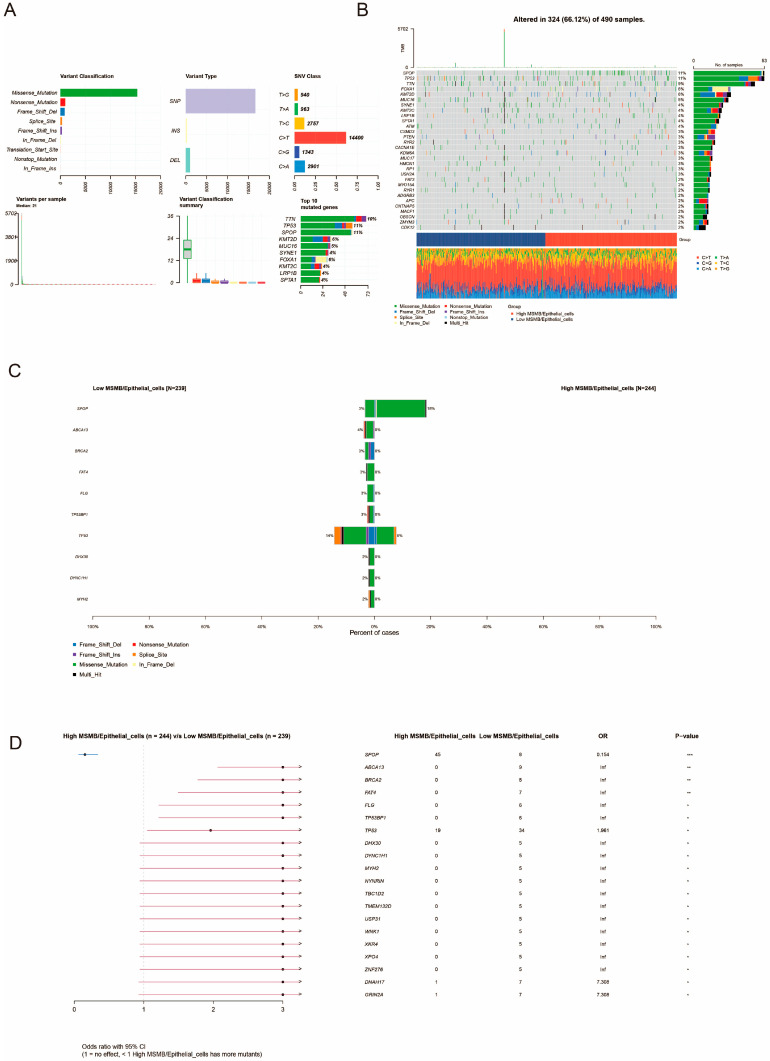
Differential analysis of genetic variation in MSMB/Epithelial_cells high- and low-ratio groups. (**A**) Summary information of PCa genetic variation. (**B**) Waterfall plots of the top 30 mutated genes between MSMB/Epithelial_cells high and low percentage groups in PCa. (**C**) Stacked barplot of the frequency difference analysis of mutant genes associated with MSMB/Epithelial_cells subgroups. (**D**) Forest plot of mutant genes associated with MSMB/Epithelial_cells subgroups. * *p* < 0.05; ** *p* < 0.01; *** *p* < 0.001.

**Figure 9 biomedicines-13-00487-f009:**
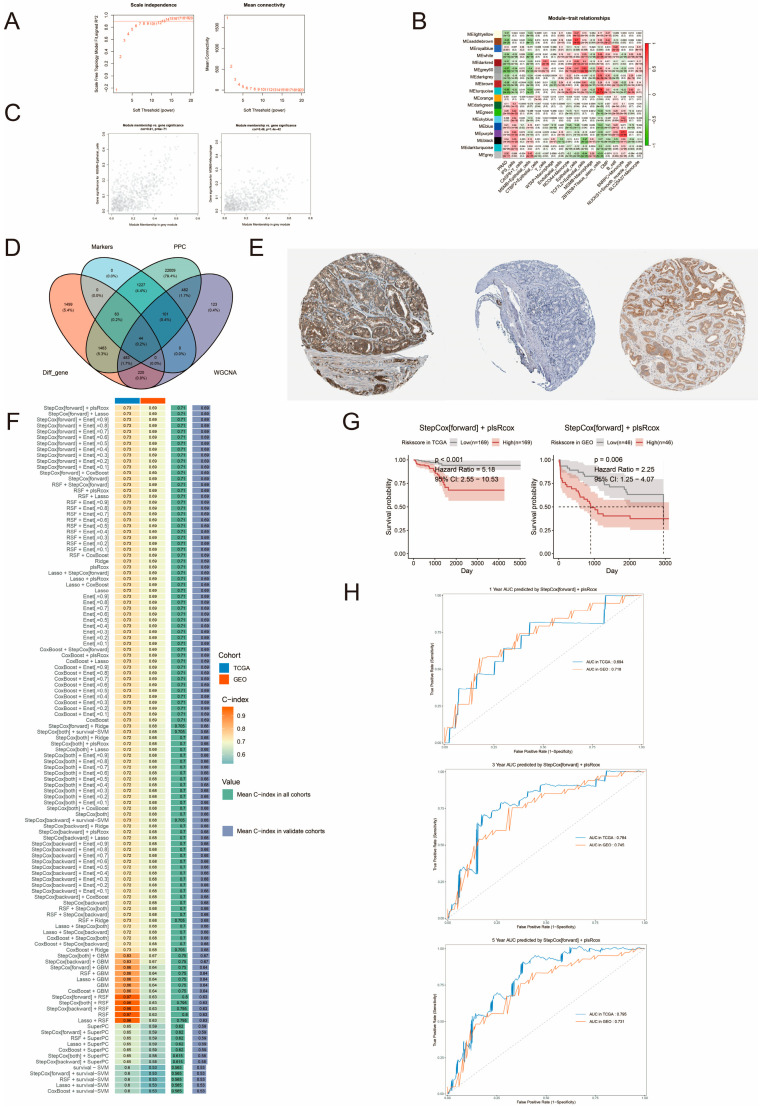
Screening of differential genes and construction of PCa DFI prognostic model. (**A**) Correlation analysis of R2 and Mean Connectivity with soft thresholds, respectively. (**B**) Heatmap of correlation analysis of different modules with convolutional cells. Red indicates a positive correlation, and green indicates a negative correlation. (**C**) Correlation scatter plots of gray modules with MSMB/Epithelial_cells and MSMB/Macrophage, respectively. (**D**) Venn diagram screening for core genes. (**E**) Images from left to right show the staining of GGCT, C2, and MRPL41 in PCa tissue. (**F**) Construction of PCa DFI prognostic model based on 117 algorithm combinations of 10 machine learning. Blue color represents TCGA database, and red color represents GEO database. The color gradient indicates the size of the C-index. Darker colors indicate larger C-index values. Green in Value indicates the mean C-index value in all cohorts, while blue indicates the mean C-index value in the validation cohort. (**G**) Survival analysis of high- and low-risk groups in the training and vali-dation sets. The red curve represents the high-risk group, and the gray curve represents the low-risk group. (**H**) ROC curves of StepCox[forward] + plsRcox algorithm at 1, 3, and 5 years in the training and validation sets. The blue curve indicates the AUC values in the TCGA database, and the red curve indicates the AUC values in the GEO database.

**Figure 10 biomedicines-13-00487-f010:**
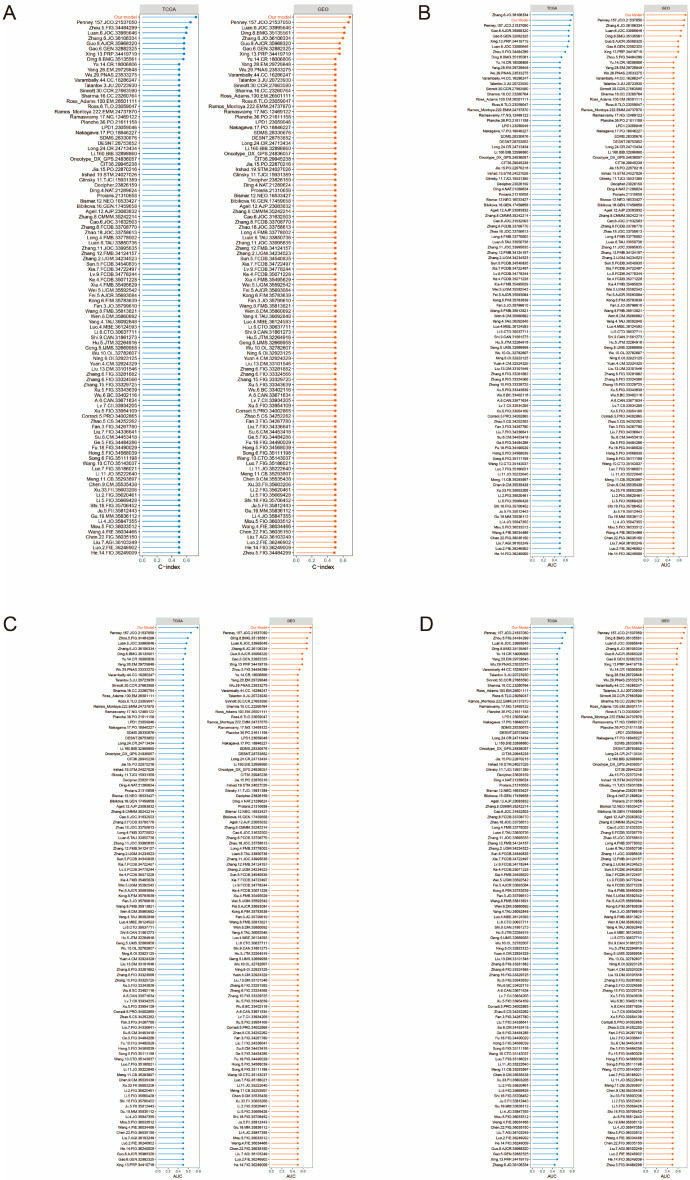
Inter-model comparison between training and validation sets. (**A**) Comparison of the C-index of our model with 102 published PCa prognostic models. (**B**–**D**) Comparison of predicted AUC values at 1, 3, and 5 years between our model and 102 published models.

**Figure 11 biomedicines-13-00487-f011:**
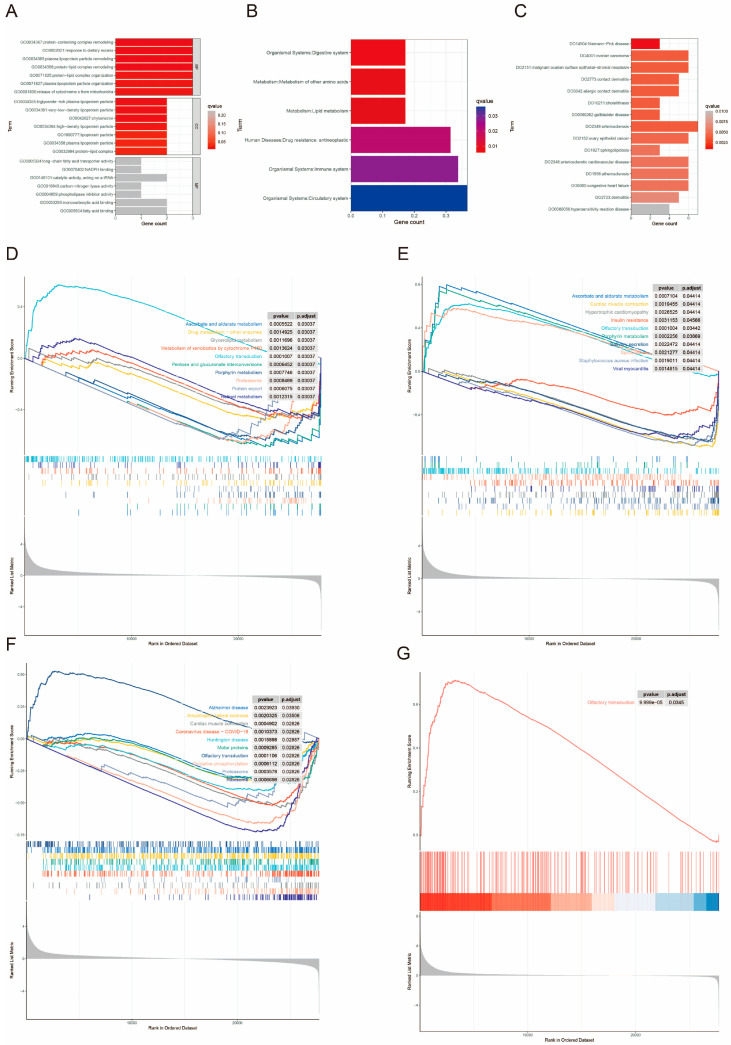
Gene enrichment analysis. (**A**) GO analysis of core genes. The color gradient indicates the magnitude of the q-value; the darker the color, the smaller the q-value. (**B**) KEGG analysis of core genes. The color gradient indicates the magnitude of the q-value; the redder the color, the smaller the q-value; the bluer the color, the larger the q-value. (**C**) DO analysis of core genes. The color gradient indicates the magnitude of the q-value; the darker the color, the smaller the q-value. (**D**–**G**) GSEA analysis of prognosis-related genes. Different colored curves represent different metabolic pathways.

**Figure 12 biomedicines-13-00487-f012:**
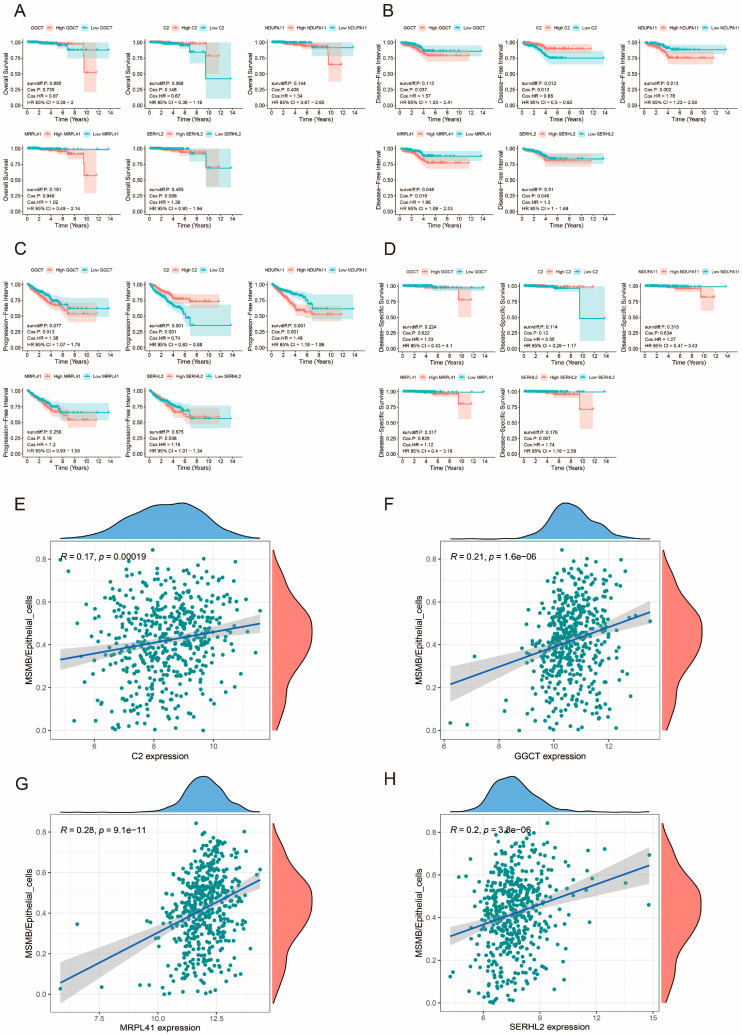
Prognostic analysis of prognosis-related genes in PCa and their correlation analysis with MSMB/Epithelial_cells. (**A**) OS survival analysis of prognosis-related genes in PCa. (**B**) DFI survival analysis of prognosis-related genes in PCa. (**C**) PFI survival analysis of prognosis-related genes in PCa. (**D**) DSS survival analysis of prognosis-related genes in PCa. The red color in Figure (**A**–**D**) represents the high-expression group of genes, while the blue color represents the low-expression group of genes. (**E**–**H**) Correlation analysis of prognosis-related genes with MSMB/Epithelial_cells. A larger R value indicates a stronger correlation, and a *p*-value less than 0.05 is considered statistically significant.

## Data Availability

No raw data were generated for this study. The sources of data used are cited in the text. All other relevant data are included in the main text.

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
