# Peer review of "A Multi-Omics-Based Exploration of the Predictive Role of MSMB in Prostate Cancer Recurrence: A Study Using Bayesian Inverse Convolution and 10 Machine Learning Combinations"

_biomedicines, 2025, doi:10.3390/biomedicines13020487_

Round 1
Reviewer 1 Report
Comments and Suggestions for Authors
This manuscript aims to develop a novel computational model to assess the predictive role of MSMB in prostate cancer. I think the authors need to trim down the amount data with clear focus. The current organization (although promising) does not help to get clear message from this manuscript. Authors should not dump all the data and hope readers can get the idea. It would be helpful if authors clearly mentions the input and output of each analysis.
Major comments
1. There are several references that are missing. Please, cite each data source and computational packages with its original paper (e.g. GEO, TCGA, WGCNA, KEGG, scPagwas, etc.).
2. Section 3.1: Although presenting the raw data on Table S1 is great, authors should make it interpretable (e.g. meaning of each column and how it was generated). Making a separate worksheet with key genes will be helpful. Table S1 suggests that MSMB is the gene with one of the lowest p values, but authors claim it is inconsistent.
3. Figure 1D: In general, the fraction of certain cell types (especially from patient samples) cannot be estimated as the samples are not collected exhaustively. The only time such estimation is meaningful is when a replicate tumors were used.
4. Figure 2: Authors need to clearly present data used here. Are all cells from one scRNA sample or from multiple samples? Also, please describe all the features in the image (what is the color, is it average in the pixel/loci etc).
5. Figure 4: What is the cutoff for the low/high population in Figure 4D? Also, are there equal number of samples for low/high populations to start with? MSMB data does not look statistically significant (Fig 4D).
6. Section 3.10: Testing multiple models with only 5 genes does not require complicated machine learning models, and a simple PCA will do the trick. Also, using a few genes does not require RNA seq and can be done with a much cheaper microarray. Authors need to show the performance with 1) one of 5 genes, 2) 5 genes only, and 3) all genes. Such analysis can justify the use of RNA-seq for prognosis.
Minor comments:
1. Reference 17-19: If authors wants to discuss the limitation, it should appear in the text. Otherwise, these should be removed.
2. Figure 1F: If authors wants to discuss the decrease, it should go through a statistical measure with significance.
Author Response
Major comments
Comments 1:There are several references that are missing. Please, cite each data source and computational packages with its original paper (e.g. GEO, TCGA, WGCNA, KEGG, scPagwas, etc.).
Response 1:Thank you for pointing this out. We agree with this comment. We cite references for TCGA, GEO, WGCNA, GO, KEGG, and DO as 18, 19, 27, 35, 36, and 37, respectively.
Comments 2:Section 3.1: Although presenting the raw data on Table S1 is great, authors should make it interpretable (e.g. meaning of each column and how it was generated). Making a separate worksheet with key genes will be helpful. Table S1 suggests that MSMB is the gene with one of the lowest p values, but authors claim it is inconsistent.
Response 2:Thank you for pointing this out. We agree with this comment. We have explained the meaning of the columns in Table S1 and placed them at the end of the table. The results of our pQTL analysis of MSMB are placed separately in Table S2. The claim in the manuscript that MSMB is inconsistent is incorrect, and we thank you for the correction. The revisions are located in red in Section 3.1.
Comments 3: Figure 1D: In general, the fraction of certain cell types (especially from patient samples) cannot be estimated as the samples are not collected exhaustively. The only time such estimation is meaningful is when a replicate tumors were used.
Response 3:Thank you for pointing this out. We agree with this comment. Due to the limitations of the current samples, it is not possible to fully capture all the heterogeneity in the tumor microenvironment, which can lead to an inability to do an accurate estimation when analyzing the proportion of a certain cell type, which can indeed lead to unreliable results. However, this analysis in this study is only a preliminary study to get a general idea of how the proportion of a certain cell type changes. As you said, in the future we will use multiple samples from the same patient or repeat biopsies to make reliable estimates of cell types.
Comments 4: Figure 2: Authors need to clearly present data used here. Are all cells from one scRNA sample or from multiple samples? Also, please describe all the features in the image (what is the color, is it average in the pixel/loci etc).
Response 4:Thank you for pointing this out. We agree with this comment. Figure 2 is based on scRNA-seq data. The cells shown in this figure are from a single sample, sample GSM6890196 in GSE221603. We have described the applied data in detail, and the revisions are highlighted in red at the end of the second paragraph in section 2.2 and in red in section 2.4. We have made detailed revisions to the legend of Figure 2 to make it easier for readers to understand the meaning of the image. The changes are highlighted in red.
Comments 5: Figure 4: What is the cutoff for the low/high population in Figure 4D? Also, are there equal number of samples for low/high populations to start with? MSMB data does not look statistically significant (Fig 4D).
Response 5:Thank you for pointing this out. We agree with this comment. The threshold for the low/high population in Figure 4D is the median gene expression value, with samples above the median categorized as high risk and vice versa as low risk. The number of high- and low-risk populations is essentially the same, close to 1:1. For MSMB related images do not appear to be very significant, probably because individual differences between samples, fluctuating gene expression values and other issues result in differences that are not easily distinguishable by visualization, but this does not prevent statistical significance.
Comments 6: Section 3.10: Testing multiple models with only 5 genes does not require complicated machine learning models, and a simple PCA will do the trick. Also, using a few genes does not require RNA seq and can be done with a much cheaper microarray. Authors need to show the performance with 1) one of 5 genes, 2) 5 genes only, and 3) all genes. Such analysis can justify the use of RNA-seq for prognosis.
Response 6:Thank you for your valuable comments, and we agree that evaluation using simple methods such as PCA, microarrays, etc. may be simpler and more optimized. However, we would like to clarify the following points: 1. The biology of cancer is extremely complex, with interactions between genes; 2. RNA-seq can provide a more complete landscape of gene expression; 3. More sophisticated models such as machine learning assess all genes in a way that provides a more realistic scenario for clinical applicability, whereas PCA may fail to capture the complexity of the relationship with clinical outcomes. Meanwhile, our study has already analyzed the C-index and AUC values of the five gene models, and the results showed better predictive ability; we also conducted a detailed analysis of the role of each gene in different prognostic endpoints (DFI, PFI, etc.), and the results showed that the different genes have certain predictive ability in prognostic endpoints, so we do not think that there is a need for further comparative analyses. We believe that the combination of 5 genes is sufficient to support the rationality of RNA-seq in prognostic analysis.
Minor comments
Comments 1: Reference 17-19: If authors wants to discuss the limitation, it should appear in the text. Otherwise, these should be removed.
Response 1:Thank you for pointing this out. We agree with this comment. We finally decided to drop the discussion of this restriction and remove the related content and documentation.
Comments 2: Figure 1F: If authors wants to discuss the decrease, it should go through a statistical measure with significance.
Response 2:Thank you for your valuable comments. In our study, we chose to preliminarily explore the differences in communication patterns between normal tissues and PCa, and between normal tissues and mHSPC, based on the observed differences in the strength and number of cellular communications. While statistical tests certainly add to the robustness of the findings, we believe that examining the quantitative relationship between cellular communication can provide important preliminary insights into the biological interactions that occur between these tissues. Although statistical significance tests could not be performed in this study, this change in trend could be verified in future follow-up studies by increasing the sample size or in other ways. Furthermore, the initial analysis of quantitative changes is of interest as the sample size of the data set is limited or there is a large variation between samples, and statistical tests may not reflect the biological signals.
Reviewer 2 Report
Comments and Suggestions for Authors
The presented paper is devoted to bioinformatic analysis of the samples from prostate cancer patients. The authors analyzed the gene expression profile and compared them with available dataset (GEO, KEGG etc) using machine learning algorithms. From the analysis the authors selected 5 genes, which could serve as prognostic markers of the post-treatment biochemical recurrence along with the major prognostic factor of prostate cancer, prostate specific antigen. The amount of bioinformatical data is sufficient for publication but there are a few comments to be addressed.
- The description of the cohorts of the patients must be broadened: the number of patients, diagnosis, resistance, metastasis, age, inclusion/exclusion criteria, treatment, type of sample (blood, cancer tissue) must be mentioned. The correlations between patient characteristics and gene expression profiles could be the alternative option to discuss.
- The 5 selected genes must be validated in routine laboratory testing (predominantly qPCR or immunohistochemistry of tumor FFPE blocks).
- The English editing should be performed: the absence of spaces, typos (“macrophages” instead “macophages” etc), long sentences difficult to read must be corrected. The text should be revised to be more concise and better organized.
- The information of candidate prognostic genes should be summarized in a table.
- The information concerning actual methods of diagnosis and treatment of prostate cancer should be added to Introduction section.
- The English editing should be performed: the absence of spaces, typos (“macrophages” instead “macophages” etc), long sentences difficult to read must be corrected. The text should be revised to be more concise and better organized.
Author Response
Comments 1:The description of the cohorts of the patients must be broadened: the number of patients, diagnosis, resistance, metastasis, age, inclusion/exclusion criteria, treatment, type of sample (blood, cancer tissue) must be mentioned. The correlations between patient characteristics and gene expression profiles could be the alternative option to discuss.
Response 1:Thank you for pointing this out. We agree with this comment. Therefore, we have described the details of the patient cohort, with the revisions in red in the first paragraph of section 2.2. The discussion of the correlation between patient characteristics and gene expression profiles that you describe is a good method of analysis, but the article's analysis was already so full that we ultimately decided not to perform that analysis. In the future, we will apply that analysis to the discussion of correlations between patient clinical characteristics and genes. Finally, thank you for your valuable suggestions.
Comments 2:The 5 selected genes must be validated in routine laboratory testing (predominantly qPCR or immunohistochemistry of tumor FFPE blocks).
Response 2:Thank you for pointing this out. We agree with this comment. Due to the limitations of the experimental conditions, we were not able to perform the correlation analysis of prognostic genes by laboratory tests. However, we extracted the immunohistochemical results of three prognostic genes from the HPA database for verification of gene staining in PCa. The revisions are in red in Section 3.10 and in Figure 9E.
Comments 3:The English editing should be performed: the absence of spaces, typos (“macrophages” instead “macophages” etc), long sentences difficult to read must be corrected. The text should be revised to be more concise and better organized.
Response 3:Thank you for pointing this out. We agree with this comment. As a result, we read through the text, corrected the absence of spaces and typos, and revised long sentences that were difficult to read, making the text more concise.
Comments 4:The information of candidate prognostic genes should be summarized in a table.
Response 4:Thank you for pointing this out. We agree with this comment. We summarized information on prognosis-related genes in Table S6, which can be found in the Supplementary file.
Comments 5:The information concerning actual methods of diagnosis and treatment of prostate cancer should be added to Introduction section.
Response 5:Thank you for pointing this out. We agree with this comment. We have added content about the diagnosis and treatment of prostate cancer to the Introduction section. The revisions are in red in that section.
Reviewer 3 Report
Comments and Suggestions for Authors
1. Many figures are overly dense and challenging to interpret. For example, the heatmaps and survival curves need more detailed legends and better scaling.
2. Justify the use of 117 algorithms. Was this exhaustive approach computationally efficient, or could a more targeted strategy have been equally effective?
3. Address why the high proportion of MSMB/Epithelial_cells correlates with better outcomes, incorporating literature on MSMB’s biological role in epithelial cells.
4. Correct grammatical errors and awkward phrasing, such as in the conclusion: "This signature can guide patients to personalized treatment for improved prognosis." Revise for fluency and precision.
Author Response
Comments 1:Many figures are overly dense and challenging to interpret. For example, the heatmaps and survival curves need more detailed legends and better scaling.
Response 1:Thank you for pointing this out. We agree with this comment.
Therefore, We have revised the legends for almost all the figures in the manuscript, aiming to more accurately describe the content of the images and make it easier for readers to understand.
Comments 2:Justify the use of 117 algorithms. Was this exhaustive approach computationally efficient, or could a more targeted strategy have been equally effective?
Response 2:Thank you for pointing this out. We agree with this comment.
We use a combination of multiple algorithms in order to enhance the credibility of the results and improve the accuracy of the predictions, since no single algorithm is guaranteed to be optimal in complex oncology problems. At the same time, using multiple algorithms allows us to combine the strengths of different algorithms to handle more complex data. Although 117 algorithms may not be the most computationally efficient method, this approach provides an opportunity to explore the possibilities and identify the best performing models
Comments 3:Address why the high proportion of MSMB/Epithelial_cells correlates with better outcomes, incorporating literature on MSMB’s biological role in epithelial cells.
Response 3:Thank you for pointing this out. We agree with this comment.
We summarize the relevant literature on the role of MSMB in prostate cancer to provide a theoretical basis for us to focus on MSMB analysis. The modifications are highlighted in red in the second paragraph of the Discussion section.
Comments 4:Correct grammatical errors and awkward phrasing, such as in the conclusion: "This signature can guide patients to personalized treatment for improved prognosis." Revise for fluency and precision.
Response 4:Thank you for pointing this out. We agree with this comment.
First, we modified the sentence “This signature can guide patients to personalized treatment for improved prognosis.” in the conclusion, and then corrected some grammatical errors and awkward phrasing throughout the text.
Reviewer 4 Report
Comments and Suggestions for Authors
The manuscript titled "A multi-omics-based exploration of the predictive role of MSMB in prostate cancer recurrence: a study using Bayesian inverse convolution and 10 machine learning combinations" provides a thorough and well-structured study on the development and validation of a novel predictive model for prostate cancer recurrence. The manuscript integrates Bayesian deconvolution, single-cell transcriptomics, and machine learning methodologies to address a critical unmet need in prostate cancer risk stratification and treatment personalization.
The study is innovative, employs state-of-the-art methodologies, and the results demonstrate strong clinical relevance. However, several areas require clarification, additional analysis, and editorial improvements for the manuscript to reach its full potential.
1.There are some grammatical errors in the manuscript. The author is advised to carefully review and make necessary corrections.
2.The abbreviations section is comprehensive, but some terms, such as TRS and plsRcox, are not immediately intuitive and require clearer explanation upon first mention.
3.While the model outperforms 102 published prognostic signatures, the manuscript does not sufficiently elaborate on the comparative limitations of these models, such as data constraints and algorithmic biases. Provide more details on the limitations of previously published models and how the current study addresses these gaps. Discuss how prospective validation could be conducted in future research.
4.The manuscript provides a detailed explanation of data preprocessing, including RNA-seq normalization and exclusion criteria for low-quality cells. However, the rationale for specific thresholds, such as mitochondrial gene proportion >20%, needs further justification, as different thresholds could impact the analysis.
5.The manuscript utilizes a variety of machine learning algorithms to enhance the predictive performance of your models. To further strengthen the credibility and scientific foundation of the manuscript, I recommend citing key bioinformatics literature, particularly those related to interpretable machine learning in bioinformatics, such as doi:10.1016/j.eswa.2023.122964L, 10.1016/j.asoc.2024.111523, 10.1186/s13059-024-03357-w, 10.1021/acs.jcim.4c01991 and so on.
Comments on the Quality of English Language
The English could be improved to more clearly express the research.
Author Response
Comments 1:There are some grammatical errors in the manuscript. The author is advised to carefully review and make necessary corrections.
Response 1:Thank you for pointing this out. We agree with this comment. Therefore, we read through the entire text and corrected some of the grammatical errors.
Comments 2:The abbreviations section is comprehensive, but some terms, such as TRS and plsRcox, are not immediately intuitive and require clearer explanation upon first mention.
Response 2:I have a slight disagreement with this comment. Both TRS and plsRcox are clearly explained in the first mention. The explanation of TRS can be seen on page 9 at line 28, and the explanation of plsRcox can be seen on page 4 at line 27.
Comments 3:While the model outperforms 102 published prognostic signatures, the manuscript does not sufficiently elaborate on the comparative limitations of these models, such as data constraints and algorithmic biases. Provide more details on the limitations of previously published models and how the current study addresses these gaps. Discuss how prospective validation could be conducted in future research.
Response 3:Thank you for pointing this out. We agree with this comment. Therefore, in the manuscript we add the strengths of this study in comparison to other studies, especially in terms of data constraints and algorithmic biases; and a discussion of how prospective validation can be performed in future studies. The revisions are shown in red in the sixth reddened paragraph of the Discussion section.
Comments 4:The manuscript provides a detailed explanation of data preprocessing, including RNA-seq normalization and exclusion criteria for low-quality cells. However, the rationale for specific thresholds, such as mitochondrial gene proportion >20%, needs further justification, as different thresholds could impact the analysis.
Response 4:Thank you for pointing this out. We agree with this comment.
Therefore, we added the significance of the mitochondrial gene proportion for screening high-quality cells and the analysis of different thresholds to the manuscript, and cited some of the literature to justify the threshold of 20%. The revised content is highlighted in red and is located in the middle section of the second paragraph of section 2.2 in the "Methods" part.
Comments 5:The manuscript utilizes a variety of machine learning algorithms to enhance the predictive performance of your models. To further strengthen the credibility and scientific foundation of the manuscript, I recommend citing key bioinformatics literature, particularly those related to interpretable machine learning in bioinformatics, such as doi:10.1016/j.eswa.2023.122964L, 10.1016/j.asoc.2024.111523, 10.1186/s13059-024-03357-w, 10.1021/acs.jcim.4c01991 and so on.
Response 5:Thank you for pointing this out. We agree with this comment.
For this purpose, we cite part of the bioinformatics literature on machine learning, as cited in 29, 30, 31. The citation has been red-flagged and is located on page 4.
Round 2
Reviewer 1 Report
Comments and Suggestions for Authors
The manuscript has improved, but I still have some concerns. Also, it would be great if authors can make high resolution figures, and a clear summary of each section would make the manuscript stronger. Right now the main problem is the link between each section. There are enough data and figures, but figuring out how each section is connected to different sections is very difficult.
1. The manuscript is still underreferenced. Please, cite technical resources and data sources properly.
2. Section 3.4: It is not clear if the authors’ claim about PFI is true. The whole section seems misleading. Abstract needs to modified as well.
3. Figure 7: Axis label is missing.
Author Response
Comments 1: The manuscript is still underreferenced. Please, cite technical resources and data sources properly.
Response 1:Thank you for pointing this out. We agree with this comment. We have cited additional literature, 20, 24, 25, 26, and 27. Both the BayesPrism package and the scPagwas package have cited relevant literature, 17 and 43, respectively.
Comments 2: Section 3.4: It is not clear if the authors’ claim about PFI is true. The whole section seems misleading. Abstract needs to modified as well.
Response 2:I have a different opinion on this matter. Section 3.4 is an analysis of the prognostic characteristics of convoluted cells, aimed at helping readers better understand the role of MSMB/Epithelial_cells subgroups in the prognosis of PCa, thus providing a basis for further analysis of this subgroup. Regarding your concern about PFI, are you referring to the issue with the figure? I addressed this point in my first revision, and I hope my explanation clarifies your doubts. I have also revised the abstract to make it more comprehensive and better help readers understand the overall content of the manuscript.
Comments 3: Figure 7: Axis label is missing.
Response 3:Thank you for pointing this out. We agree with this comment. The reason why the figures A and C in Figure 7 are not labeled with horizontal coordinates is because the significance of the horizontal coordinates is the grouping information, and the groupings have been labeled at the top of the figures.
Reviewer 2 Report
Comments and Suggestions for Authors
No further comments
Author Response
Thank you for taking the time to review my manuscript and for your constructive comments.
Reviewer 3 Report
Comments and Suggestions for Authors
The biggest issue still lies in the figure quality and the representation of their work. The authors addressed all my comments. But there are things that they must improve to make it a successful manuscript.
1. Figure quality needs to be taken care seriously. In Fig. 5, all the legends are not readable at all. Please try to upload the original files. I suggest the authors to preview the uploaded file. Not sure if it is system issue or simply the original files issue.
2. For the methods part, they addressed all my previous comments. A new comment from me, that will make this manuscript more successful is that, to include a brief discussion of computational cost and potential trade-offs in model selection.
3. I did notice the improvement of writing but there are still improvements can be made. Some figure captions remain lengthy and overly detailed. Simplifying descriptions while keeping essential information would improve readability.
Once the authors address all my comments well, this manuscript is good in its shape.
Comments on the Quality of English LanguageThe biggest issue still lies in the figure quality and the representation of their work.
Author Response
Comments 1:Figure quality needs to be taken care seriously. In Fig. 5, all the legends are not readable at all. Please try to upload the original files. I suggest the authors to preview the uploaded file. Not sure if it is system issue or simply the original files issue.
Response 1:Thank you for pointing this out. We agree with this comment. We have checked the uploaded files and did not find any problems with Figure 5, we re-uploaded the zip of the figures and hopefully it will match the quality of the figures in the journal.
Comments 2:For the methods part, they addressed all my previous comments. A new comment from me, that will make this manuscript more successful is that, to include a brief discussion of computational cost and potential trade-offs in model selection.
Response 2:Thank you for pointing this out. We agree with this comment. We therefore conduct a discussion of computational costs and model selection, with the revisions located in the second paragraph of Section 3.10 marked in red.
Comments 3:I did notice the improvement of writing but there are still improvements can be made. Some figure captions remain lengthy and overly detailed. Simplifying descriptions while keeping essential information would improve readability.
Response 3:Thank you for pointing this out. We agree with this comment. We have modified the figure captions to simplify them as much as possible and retain the basic information to improve readability.
Reviewer 4 Report
Comments and Suggestions for Authors
N/A
Author Response

(The authors gave the same response as above.)

Round 3
Reviewer 1 Report
Comments and Suggestions for Authors
Accept as it is.